# A Self-Attention-Based Imputation Technique for Enhancing Tabular Data Quality

**Do-Hoon Lee and Han-joon Kim *** 

School of Electrical and Computer Engineering, University of Seoul, 163 Seoulsiripdaero,
Seoul 02504, Republic of Korea
* Correspondence: khj@uos.ac.kr; Tel.: +82-10-5054-5202

**Abstract:** Recently, data-driven decision-making has attracted great interest; this requires high-quality datasets. However, real-world datasets often feature missing values for unknown or intentional reasons, rendering data-driven decision-making inaccurate. If a machine learning model is trained using incomplete datasets with missing values, the inferred results may be biased. In this case, a commonly used technique is the missing value imputation (MVI), which fills missing data with possible values estimated based on observed values. Various data imputation methods using machine learning, statistical inference, and relational database theories have been developed. Among them, conventional machine learning based imputation methods that handle tabular data can deal with only numerical columns or are time-consuming and cumbersome because they create an individualized predictive model for each column. Therefore, we have developed a novel imputational neural network that we term the Denoising Self-Attention Network (DSAN). Our proposed DSAN can deal with tabular datasets containing both numerical and categorical columns; it considers discretized numerical values as categorical values for embedding and self-attention layers. Furthermore, the DSAN learns robust feature expression vectors by combining self-attention and denoising techniques, and can predict multiple, appropriate substituted values simultaneously (via multi-task learning). To verify the validity of the method, we performed data imputation experiments after arbitrarily generating missing values for several real-world tabular datasets. We evaluated both imputational and downstream task performances, and we have seen that the DSAN outperformed the other models, especially in terms of category variable imputation.

**Keywords:** attention network; deep learning; multi-task learning; embedding; data quality; missing values; data imputation

## 1. Introduction

As the amount of data increases, data-driven decision-making via machine learning has become increasingly important in many fields. The results are greatly affected by data quality; good quality data are essential for data-driven decision-making. The most common problem is missing values, which damage the data pipelines and render the results of data-driven tasks inaccurate. If a machine learning model is trained using incomplete datasets, the inferred results may be biased [1]. For example, many missing values arise because customers do not wish to give personal information when asked to comment on e-commerce or mobile advertising domains. In such cases, data scientists or machine learning engineers must clean the datasets to ensure meaningful results. Missing value imputation (MVI) is the most commonly adopted solution. MVI seeks to replace the missing values with substituted values estimated based on the observed values [2].

In general, MVI methods use a statistical or machine learning technique to replace missing values with substituted values. Statistical MVI methods employ descriptive statistics such as mean and mode values; they are thus simple and fast, but inaccurate. Machine learning based MVI methods have recently been studied to ensure more accurate

results. Such methods generate or predict substituted values using trained generative or predictive models. MVI methods employing generative models include Generative Adversarial Imputation Nets (GAIN) [3] and denoising autoencoder (DAE) for multiple imputations [4]; MVI methods that feature predictive models include MissForest [5] and Datawig [6]. Generative-model-based MVI methods cannot generate substituted values for categorical variables (or columns); such models cannot handle tabular data that contain both numerical and categorical variables. Meanwhile, predictive-model-based MVI methods train models for each variable, and can thus deal with tabular data [7]. However, it is both cumbersome and inefficient to train several predictive models.

Here, we have developed a novel, end-to-end, imputational neural network termed the Denoising Self-Attention Network (DSAN), which can handle tabular data with heterogeneous columns. The DSAN combines self-attention-based feature representation and denoising to learn robust features prior to analysis of an incomplete mixed-type dataset. The DSAN is trained by using a multi-task learning (MTL) method to predict substituted values that are appropriate for the type of variable under consideration. DSAN training proceeds in a self-supervised manner; each received input is re-predicted. During such training, the DSAN learns the interactions among variables and among observed and missing values. Thus, the DSAN predicts substituted values appropriate for each type of variable, and missing values.

To verify the method, we performed experiments evaluating two types of performance: imputation performance and downstream task performance. Imputation performance is evaluated with the accuracy by which a method imputes a substituted value, and is calculated as the errors between the imputed and original values. Downstream task performance explores whether the imputed dataset can be used to solve other tasks. In order to explore downstream task performance, a binary classifier was trained using an imputed dataset, and its performance is then evaluated.

The rest of the paper is organized as follows. Section 2 describes the related work and the differences between our method and existing methods. In Section 3, we formalize the data imputation problem. Section 4 introduces our DSAN imputation method in detail. Section 5 describes our experimental setup and results. Lastly, we conclude our paper in Section 6.

## 2. Related Work

For enhancing tabular data quality, besides MVI, data repairing is used to update any detected erroneous data with correct data. For this, there has been a study that probabilistically calculates the correct value for erroneous data through statistical inference after detecting erroneous data using integrity constraints [8]. As a study on solving the MVI problem using integrity constraints, Breve et al. [9] proposed a method to impute missing values by identifying candidate values that ensure data integrity using relaxed functional dependencies. Similarly, Song et al. [10] showed that data imputation is possible with valid candidate values (called neighbors) that can be identified by a similarity relationship. Furthermore, Jia et al. [11] devised tensor-based data imputation models to extract latent features from a traffic dataset and predict neighbors to fill in the missing values via tensor factorization.

Recently, deep neural network-based imputational methods for tabular data have received much attention. Deep neural networks afford several advantages, including end-to-end learning and multi-task learning (MTL), possible fusion of multiple modalities (e.g., images and text), and representation learning. Efforts have been made to apply such models to MVI tasks. The representational learning of a deep neural network is used to extract features of tabular data that aid predictive imputation [6,12,13]. MTL can simultaneously handle regression and classification tasks; thus, an imputational model can combine a regression task and a classification task [14,15].

The *attention* mechanism employed in machine translation [16] has been successfully ported to natural language processing tasks [17,18] and computer vision [19]. A number of

attempts have been made to incorporate the mechanism into deep neural networks that handle tabular data. TabNet [17] used an *attention* mechanism to imitate a decision tree, which works well when used to handle structured data. TabNet enabled interpretable and efficient learning; sequential *attention* was used to select the important features of each decision-making stage. TabTransformer [20] used the encoder of a transformer [21] and a self-attention mechanism to map categorical features into a contextual embedment. TabTransformer outperformed a multi-layer perceptron (MLP) and a tree-based ensemble model, and was very robust against noisy and missing data.

Our imputation method has a special deep learning architecture that is quite different from the existing methods mentioned above.

- Our method has a self-attention neural network architecture to realize effective representation learning for noisy data.
- We use MTL to predict missing values simultaneously for various column types in the imputation task, so as not to combine the imputation task and downstream task.
- Generally, the *attention*-based neural networks for tabular data [17,20] have been proposed to solve the classification problem. In contrast, our method includes the *attention* mechanism to solve the missing value imputation problem.
- Our method considers discretized numerical values as categorical values for embedding and self-attention layers.

## 3. Problem Formulation

We seek to transform an incomplete tabular dataset (e.g., Figure 1) into a complete dataset (e.g., Figure 2) by predicting substitute values that impute missing values. The tabular data of interest include both numerical and categorical variables; we solve regression and classification problems simultaneously.

| age | job | martial | education | default | ... | duration | campaign | pdays | poutcome | deposit |
|-----|-----|---------|-----------|---------|-----|----------|----------|-------|----------|---------|
| 59 | admin. | married | secondary | no | ... | 1042 | 1 | NaN | 0 | yes |
| 56 | admin. | married | secondary | no | ... | 1467 | 1 | −1 | 0 | yes |
| 41 | technician | married | secondary | no | ... | NaN | 1 | −1 | 0 | yes |
| 55 | services | married | secondary | no | ... | 579 | 1 | −1 | 0 | yes |
| 54 | admin. | NaN | NaN | no | ... | 673 | 2 | −1 | 0 | yes |

**Figure 1.** Example of an input dataset with incomplete tabular data. This is the Bank dataset used for performance evaluation in Section 6.

| age | job | martial | education | default | ... | duration | campaign | pdays | poutcome | deposit |
|-----|-----|---------|-----------|---------|-----|----------|----------|-------|----------|---------|
| 59 | admin. | married | secondary | no | ... | 1042 | 1 | −1 | 0 | yes |
| 56 | admin. | married | secondary | no | ... | 1467 | 1 | −1 | 0 | yes |
| 41 | technician | married | secondary | no | ... | 1389 | 1 | −1 | 0 | yes |
| 55 | services | married | secondary | no | ... | 579 | 1 | −1 | 0 | yes |
| 54 | admin. | married | tertiary | no | ... | 673 | 2 | −1 | 0 | yes |

**Figure 2.** Example of the output data (imputed data) that are sought.

Let $x = \{x^{num}, x^{cat}\}$ denote an input vector (a data record). An input vector $x$ contains $n$ numerical variables $x^{num} \in \mathbb{R}^n$ and $k$ categorical variables $x^{cat} = \{x_1^{cat}, x_2^{cat}, \cdots, x_k^{cat}\}$. The $i$-th categorical variable $x_i^{cat}$ has a domain set $C_i$ in which all observed values of the $i$-th categorical variable $x_i^{cat}$ lie. We use an indicator vector $m = \{m^{num}, m^{cat}\} \in \{0,1\}^{(n+k)}$ to indicate whether a given variable is missing or not. The indicator value $m_i = 1$ if the corresponding $i$-th variable (numerical or categorical) is observed; otherwise, $m_i = 0$. These values are later used to compute the loss function (employing only observed values).

Given an input vector $x$, the imputational model predicts $\hat{x}$ using the input values. We then fill in the missing values with predicted values. As mentioned above, if a variable is

numerical, we reconstruct an observed value; if a variable is categorical, we classify it by targeting the observed value.

## 4. Proposed Method

In this paper, we aim to develop a new MVI neural network model that simultaneously imputes missing values present in mixed-type tabular data containing both numeric and categorical columns. To this end, the MVI neural network must extract significant data distribution patterns from incomplete data inputs by distinguishing between numeric and categorical columns, and it should be designed with a structure that can simultaneously perform regression and classification for numerical value prediction and categorical value prediction, respectively. Taking this into account, our DSAN model features end-to-end MVI of tabular data, as shown in Figure 3. There are three modules: a feature representation module, a shared-layer module, and a task-specific module. The first module column-embeds each variable [20] and learns contextual embedding employing the self-attention layers. The model then is trained (via MTL) to predict appropriate substituted values for the different variables at the same time. The DSAN features hard parameter sharing [22]; the shared-layer module learns the shared parameters by sharing the hidden layers among all tasks. The task-specific layer module learns the various parameters that should be retained in the several task-specific output layers. The DSAN is trained in a self-supervised manner; observed values are reconstructed in a manner similar to that of AutoEncoder [23].

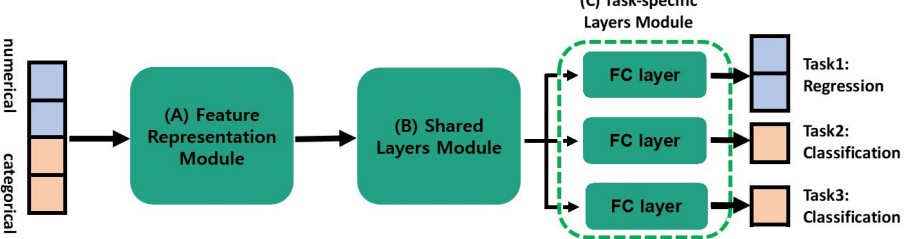

**Figure 3.** An overview of the proposed model.

### 4.1. The Feature Representation Module

The feature representation module learns how to extract useful information from incomplete mixed-type inputs, as shown as Figure 4. In this section, we explain the details.

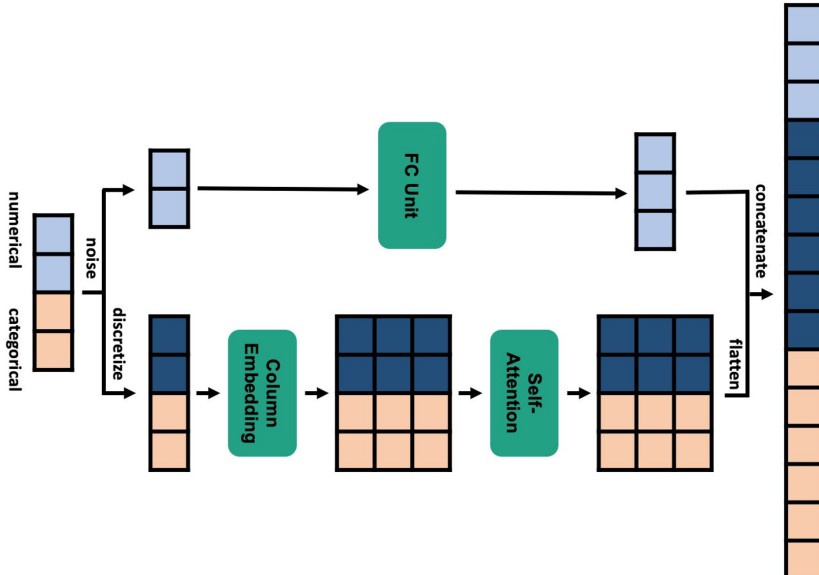

**Figure 4.** An overview of the feature representation module.

### 4.1.1. Preprocessing

To handle incomplete mixed-type data, we preprocess the input vectors as shown in Figure 5. First, we initialize a missing value (e.g., NaN) to a processible value. A missing numerical variable is initialized to 0; a missing categorical variable is initialized to a unique string, indicating that the value is missing. For example, if the value for the third column (a categorical variable) is missing, it is initialized to 'Col3:NULL'. Then, we use a denoising technique [24] to learn robust feature representation parameters for incomplete inputs. To this end, a certain percentage $\alpha$ of the observed values are randomly selected and dropped out. The dropped-out values are initialized in the manner described above, but the values of all indicator vectors $m$ are held at 1. We thus obtain corrupted input vectors $\tilde{x}$ that are input into the model.

**Figure 5.** An example of model inputs. The blue columns represent numerical columns and the brown columns represent category columns, respectively.

### 4.1.2. Column Embedding

We embed all inputs into a parametric embedment of dimension $d$ using the embedding layer. Here, the numerical variables $\tilde{x}^{num}$ are discretized (to reduce computation and ensure regularization). Given an input vector $\{\tilde{x}^{num}, \tilde{x}^{cat}\}$, an embedding matrix $E \in \mathbb{R}^{(n+k) \times d}$ is computed using

$$E = Embedding(Discretize(\tilde{x}^{num}), \tilde{x}^{cat}) \tag{1}$$

where the *Embedding* maps each value to a $d$-dimensional vector. *Discretize* converts continuous values to discrete values using certain rules in a heuristic manner. Our rules are as follows:

$$Discretize(x) = \text{int}(\log_2(x)^2) \tag{2}$$

At this time, the DSAN also learns an embedding vector with a unique value that indicates that a value is missing (e.g., 'Col3:NULL'). The information loss caused by missing values is supplemented, and feature representations for the various missing patterns (arbitrarily generated via denoising) are learned. This improves performance; the technique is similar to data augmentation [25]. The embedding matrix $E$ is input to the self-attention layer.

Some information loss is inevitable when discretizing numerical variables. To compensate for this, numerical values that are not discretized are learned in parallel via a fully connected (FC) unit consisting of an FC layer, Layer Normalization [26], the ReLU function, and Dropout. This unit is also active in the subsequent shared-layer module. In the feature representation module, the FC unit employs $d$ nodes as the embedding dimension (to unify feature size). This maintains information on the raw numerical variables, and serves as an input to the shared-layer module, along with the contextual embedding matrix computed via self-attention.

### 4.1.3. The Self-Attention Layer

As MVI predicts substituted values based on observed values, it is important to learn the relationships between the variables. Therefore, we trained the DSAN to learn the contextual embedments (including the associations) between the variables. We used self-attention to this end; this combines queries, keys, and values into single inputs and engages in feature representation by computing the relationships between elements of the input. We employed multi-head attention [21]; this linearly projects queries, keys, and values $h$ times but with different parametric weights, and (in parallel) runs the *attention* function $h$ times. It is thus possible to simultaneously focus on information from subspaces featuring different representations. Given an embedding matrix $E$, the contextual embedding matrix $H \in \mathbb{R}^{(n+k) \times d}$ is computed as follows:

$$H = Concat(\text{head}_1, \cdots, \text{head}_h)W^O \tag{3}$$

where $\text{head}_i \in \mathbb{R}^{(n+k) \times (d/h)}$ is the partial *attention* result, thus that computed by

$$\text{head}_i = Attention(EW_i^Q, EW_i^K, EW_i^V). \tag{4}$$

where the projections are parametric matrices $W_i^Q, W_i^K, W_i^V \in \mathbb{R}^{d \times (d/h)}$, and $W^O \in \mathbb{R}^{d \times d}$.

*Attention* is the scaled, dot-product *attention* function:

$$Attention(Q, K, V) = \text{softmax}(\frac{QK^T}{\sqrt{d/h}})V \tag{5}$$

The computed, contextual embedding matrix is flattened into a vector and concatenated with the output of the FC unit. Thus, we obtain a concatenated vector $z_{feature} \in \mathbb{R}^{(n+k+1) \times d}$ that is input into the shared-layer module:

$$z_{feature} = Concat(Flatten(H), FCunit(\tilde{x}^{num})) \tag{6}$$

### 4.2. The Shared-Layer and Task-Specific Layer Modules

As mentioned above, the DSAN performs regression and classification tasks in parallel using MTL. Each task is a prediction appropriate for the type of variable. If the variable is numerical, the task is a regression; if the variable is categorical, the task is a classification. We used hard parameter sharing (the most common form of MTL); this reduces overfitting by sharing the hidden layers. The components output by the feature representation module are divided between the shared-layer and the task-specific layer modules.

We created the shared-layer module $f_{share}$ by stacking FC units in layers. All units featured the same number of nodes as the dimensions of the concatenated vectors $z_{feature}$'s dimension $(n + k + 1) \times d$. We used the residual connection approach [27] to enable deep layer learning. Given $z_{feature}$, we computed the output of the shared-layer module $z_{share} \in \mathbb{R}^{(n+k+1) \times d}$ using the inputs of the several task layers:

$$z_{share} = f_{share}(z_{feature}) \tag{7}$$

The task-specific layer module featured as many FC layers as the number of tasks. During regression, the DSAN also predicts numerical variables. During classification, the DSAN predicts categorical variables separately. Thus, employing $(1 + k)$ FC layers $\{f_{num}, f_1, \cdots, f_k\}$, the DSAN is trained in a self-supervised manner; the observed input is re-predicted. The regression task FC layer $f_{num}$ reconstructs the input numerical values

$$\hat{x}^{num} = f_{num}(z_{share}) \tag{8}$$

and the other $k$ classification task FC layers $f_i$ classify the input categorical values

$$\hat{\boldsymbol{y}}_i = \sigma(f_i(\boldsymbol{z}_{share})) \tag{9}$$

where $\hat{\boldsymbol{y}}_i$ is the probability vector that the target is the $i$-th categorical value $x_i^{cat}$. If the size of the domain set of $x_i^{cat}$ is $|C_i| = 2$, this is a binary classification task, and the output $\hat{\boldsymbol{y}}_i$ is thus a scalar. The $\sigma$ activation function maps this to a probability. If a task is a binary classification, $\sigma$ is a sigmoid function; if a task features multi-class classification, $\sigma$ is a softmax function. During MTL, the total loss $\mathcal{L}_{tot}$ is the sum of the losses of each task:

$$\mathcal{L}_{tot} = \mathcal{L}_{num} + \sum_{i=1}^{k} \mathcal{L}_i \tag{10}$$

$\mathcal{L}_{num}$ is the regression task loss; the mean squared errors are computed using only the observed values:

$$\mathcal{L}_{num} = \frac{1}{n} \sum_{i=1}^{n} m_i^{num}(x_i^{num} - \hat{x}_i^{num})^2 \tag{11}$$

$\mathcal{L}_i$ is the $i$-th classification task loss; the cross-entropy loss is computed using only the observed values:

$$\mathcal{L}_i = -m_i^{cat} \sum_{j=1}^{|C_i|} y_{i,j} \log(\hat{y}_{i,j}) \tag{12}$$

We minimized the total loss $\mathcal{L}_{tot}$; all DSAN parameters are learnt in an end-to-end manner using the gradient descent method.

## 5. Experiments

### 5.1. Datasets and Evaluation Methods

We evaluated DSAN performance using several tabular datasets from the UC Irvine Machine Learning Repository [28]. Table 1 contains the details. All datasets feature several numerical and categorical variables; all contain over 10,000 records. In the real world, it is difficult to know the correct missing values; therefore, we removed all existing missing values. We then generated missing values that we knew were correct.

We evaluated two types of performance, of which the first was imputation performance assessed by calculating the errors between the imputed and original values. For numerical variables, we calculated the normalized root mean squared errors (NRMSEs) [29]; for categorical variables, we derived error rates. The second performance type was downstream task performance. We evaluated the performances of models trained with imputed datasets. The downstream task was a binary classification task; performance was evaluated by deriving area under the curve receiver operating characteristic (AUC-ROC) scores. We used the same classifier (logistic regression) for all cases.

We split the data into 80% training and 20% test sets, and created 5∼20% "missingness" in the training set using the MCAR (Missing Completely At Random) approach [30]. All imputation methods (the DSAN and the others) were compared in terms of imputations made for the incomplete training set; we then evaluated imputation performance. The classifier was trained with the imputed training sets; we then evaluated downstream task performance. To generalize the results, we applied $k$-fold cross-validation.

**Table 1.** The experimental datasets.

| Datasets | The Number of Numerical Variables | The Number of Categorical Variables | The Number of Records |
|---|---|---|---|
| Adult | 9 | 6 | 30,162 |
| Bank | 10 | 7 | 11,162 |
| Online | 10 | 8 | 12,330 |
| Churn | 5 | 6 | 10,000 |

*5.2. Experimental Settings*

We compared DSAN performance to that of a statistical (Mean/Mode) method and that of a classical machine learning imputation method (i.e., MissForest) [5] and a recent multi-task deep learning-based imputation method (i.e., MIDASpy) [15]. In the feature representation module, the DSAN featured $d = 16$ column-embedding; the multi-head attention layer can have 2~16 heads. As mentioned above, the FC unit of the feature representation module has 16 nodes, the same number as that of the embedding dimension $d$. We dropped out 40% of the input values when denoising. The shared-layer module featured six layers (FC units), each of which has $(n + k + 1) \cdot d$ nodes equal to the number of dimensions of the input feature vector. The task-specific module has an FC layer with $n$ nodes for numerical variable prediction and an FC layer for categorical variable prediction. Here, the latter includes output nodes corresponding to the domain (i.e., a set of possible values) of each categorical variable; in particular, the binary classification task is implemented as an FC layer with a single node. To create an imputation model, we trained the DSAN using the Adam optimizer [31] with $\beta_1 = 0.9$, $\beta_2 = 0.999$, $\epsilon = 10^{-8}$, learning rate $\gamma = 0.003$, and weight decay $\lambda = 10^{-5}$.

*5.3. Imputation Performance*

Figure 6 shows an example of performing imputation on the Adult dataset with the imputation model generated by the DSAN technique. Assuming that (categorical and numeric) values in the shaded cells in Figure 6 are missing according to the MCAR type, Figure 7 shows the result of imputing the missing values through the DSAN imputation model. As seen in the figure, most of missing values were replaced with correct values.

N: Numerical    C: Categorical

**Figure 6.** Part of the Adult dataset with missing values (denoted as the shaded cells).

| N | C | C | N | C | C | C | N | C | C |
|---|---|---|---|---|---|---|---|---|---|
| age | workclass | education | education.num | relationship | race | sex | hours.per.week | native.country | income |
| 82 | 0 | ⓪ | 9 | ⓪ | 0 | 0 | 18 | 0 | ⓪ |
| 54 | 0 | 1 | 4 | 1 | 0 | 0 | 40 | 0 | 0 |
| 41 | 0 | 2 | 10 | 2 | ⓪ | ⓪ | △32 | ⓪ | ⓪ |
| 34 | 0 | 0 | 9 | ① | 0 | 0 | 45 | ⓪ | ⓪ |
| 38 | 0 | 3 | 6 | 1 | ⓪ | △0 | 40 | 0 | 0 |
| 68 | 2 | 0 | [10] | ⓪ | ⓪ | 0 | 40 | 0 | 0 |
| 38 | 3 | 5 | 15 | 0 | 0 | 1 | 45 | 0 | 1 |
| 52 | 0 | 6 | 13 | 0 | 0 | 0 | 20 | ⓪ | 1 |
| 32 | 0 | ⑦ | 14 | 0 | 0 | ① | 55 | 0 | 1 |
| ㊻ | 0 | ⑤ | 15 | 0 | ⓪ | 1 | 40 | ⓪ | 1 |
| [41] | 0 | ⑧ | 7 | ⓪ | 0 | 1 | 76 | ⓪ | 1 |

N: Numerical　　　C: Categorical　　　　○ : correct　　▢ : Not bad　　△ : incorrect

**Figure 7.** An imputation result for the Adult dataset.

Table 2 shows the numerical variable imputations; we used *NRMSE* as the evaluation metric.

$$NRMSE = \sqrt{\frac{\mathbb{E}[(x - \hat{x})^2]}{Var(x)}} \tag{13}$$

where $x$ is the true and $\hat{x}$ the imputed value. The *NRMSE* is the average of the *n* numerical variables using only the missing values. In general, as the missing rate increases, imputational performance decreases because fewer observed values are used to predict substituted values. If the model is perfectly imputational, $NRMSE = 0$; if the model imputes the average value for each variable, $NRMSE \approx 1$. Therefore, a lower *NRMSE* value indicates a better imputation.

**Table 2.** Numerical variable imputational performances (NRMSEs).

| Dataset | Methods | 5% Missing | 10% Missing | 15% Missing | 20% Missing |
|---|---|---|---|---|---|
| Adult | Mean | 1.0017 | 0.9972 | 1.0002 | 1.0125 |
| | MissForest | 0.8358 | 0.8561 | 0.8745 | 0.9071 |
| | MIDASpy | 0.4420 | 0.4523 | 0.4465 | 0.4365 |
| | DSAN | 0.7999 | 0.8099 | 0.8247 | 0.8488 |
| Bank | Mean | 0.9562 | 0.9651 | 1.0000 | 0.9778 |
| | MissForest | 0.7703 | 0.8164 | 0.8586 | 0.8783 |
| | MIDASpy | 0.5027 | 0.5226 | 0.5322 | 0.6094 |
| | DSAN | 0.7580 | 0.7860 | 0.8282 | 0.8226 |
| Online | Mean | 1.0012 | 0.9836 | 0.9875 | 0.9947 |
| | MissForest | 0.6038 | 0.6504 | 0.6915 | 0.7086 |
| | MIDASpy | 0.3186 | 0.3205 | 0.3268 | 0.3498 |
| | DSAN | 0.6311 | 0.6413 | 0.6619 | 0.6738 |
| Churn | Mean | 0.9949 | 1.0052 | 0.9966 | 1.0014 |
| | MissForest | 0.9850 | 0.9964 | 0.9986 | 1.0208 |
| | MIDASpy | 0.4217 | 0.4409 | 0.4332 | 0.4233 |
| | DSAN | 0.9625 | 0.9725 | 0.9685 | 0.9757 |

Table 3 shows the average *NRMSE* values. The DSAN outperformed all other methods except MissForest in terms of numerical variable imputation. Compared to mean imputation, the performance improvements were 22.18% for the Adult, 22.05% for the Bank, 52.10% for the Online, and 3.07% for the Churn datasets. Compared to MissForest, the performance improvements were 5.79% for the Adult, 4.03% for the Bank, 1.77% for the Online, and 3.13% for the Churn datasets. For the latter dataset, we found that machine learning based regression imputation performance was rather poor when numerical variable data exhibit a uniform distribution. MIDASpy showed excellent performance in numerical variable imputation; however, it showed very poor performance in categorical variable imputation.

**Table 3.** Average numerical variable imputational performances.

| Dataset | Methods | Average | Improvement |
|---|---|---|---|
| Adult | Mean | 1.0029 | +22.18% |
| | MissForest | 0.8684 | +5.79% |
| | MIDASpy | 0.3290 | −59.92% |
| | DSAN | 0.8208 | |
| Bank | Mean | 0.9748 | +22.05% |
| | MissForest | 0.8309 | +4.03% |
| | MIDASpy | 0.5417 | −32.18% |
| | DSAN | 0.7987 | |
| Online | Mean | 0.9918 | +52.10% |
| | MissForest | 0.6636 | +1.77% |
| | MIDASpy | 0.3289 | −49.56% |
| | DSAN | 0.6520 | |
| Churn | Mean | 0.9995 | +3.07% |
| | MissForest | 1.0002 | +3.13% |
| | MIDASpy | 0.4298 | −55.68% |
| | DSAN | 0.9698 | |

Table 4 shows the results of imputation in terms of categorical variables. We used *error rate* as the evaluation metric.

$$error\ rate = \frac{|x \neq \hat{x}|}{|x|} \tag{14}$$

where $x$ is the true and $\hat{x}$ the imputed value. As for *NRMSE*, the *error rate* is the average of $k$ categorical variables using only the missing values. If the model is perfectly imputational, *error rate* = 0. Thus, a lower *error rate* indicates better imputational performance.

**Table 4.** Categorical variable imputation performances (error rates).

| Dataset | Methods | 5% Missing | 10% Missing | 15% Missing | 20% Missing |
|---|---|---|---|---|---|
| Adult | Mode | 0.4112 | 0.4129 | 0.4134 | 0.4126 |
| | MissForest | 0.2147 | 0.2238 | 0.2342 | 0.2425 |
| | MIDASpy | 0.5057 | 0.5098 | 0.5361 | 0.5385 |
| | DSAN | 0.2103 | 0.2173 | 0.2239 | 0.2325 |
| Bank | Mode | 0.4063 | 0.4088 | 0.4093 | 0.4074 |
| | MissForest | 0.2460 | 0.2558 | 0.2674 | 0.2730 |
| | MIDASpy | 0.4335 | 0.4427 | 0.4491 | 0.4356 |
| | DSAN | 0.2501 | 0.2571 | 0.2678 | 0.2732 |
| Online | Mode | 0.4166 | 0.4198 | 0.4170 | 0.4186 |
| | MissForest | 0.3387 | 0.3473 | 0.3539 | 0.3604 |
| | MIDASpy | 0.6563 | 0.6455 | 0.6701 | 0.6665 |
| | DSAN | 0.3366 | 0.3465 | 0.3498 | 0.3569 |
| Churn | Mode | 0.4055 | 0.4060 | 0.4008 | 0.4047 |
| | MissForest | 0.3561 | 0.3608 | 0.3652 | 0.3770 |
| | MIDASpy | 0.7101 | 0.7205 | 0.7293 | 0.7288 |
| | DSAN | 0.3436 | 0.3576 | 0.3549 | 0.3665 |

Table 5 shows the average error rate. Compared to mode imputation, the performance improvements were 86.66% for the Adult, 55.68% for the Bank, 20.31% for the Online, and 13.67% for the Churn datasets. Compared to MissForest, the improvements were 3.53% for the Adult, 0.76% for the Online, and 2.57% for the Churn datasets. Compared to MIDASpy, the improvements were 136.42% for the Adult, 68.02% for the Bank, 89.81% for the Online, and 63.98% for the Churn datasets. Thus, the DSAN outperformed the other models in terms of category variable imputation. Whereas MIDASpy has better numerical variable

imputation performance than the other three methods (as mentioned earlier), it showed significantly lower categorical variable imputation performance.

In terms of numerical variable imputation, the performance of our DSAN is inferior to that of MIDASpy. However, MIDASpy focuses only on improving the imputation performance for numerical variables, and in fact it does not achieve simultaneous data imputation for a given table through multi-task learning at all. In general, it is not easy to simultaneously increase the performance of learning models for all tasks in the multi-task learning. That is, the simultaneous learning of multiple tasks introduces the destructive interference problem where increasing the performance of a model on one task can degrade the performance of models on other tasks with different requirements. Although the current performance of DSAN needs to be further improved, we argue that the imputation performance for both numerical and categorical variables is reasonably balanced compared to the other imputation methods.

In addition, it is necessary to see how varying the hyperparameters (such as the number of heads and the noise ratio) inherent in our DSAN method affects the imputation performance. Figure 8 demonstrates the changes of imputation performance (i.e., *NRMSE* and *error rate*) from varying the *number of heads* of multi-head attention in the DSAN. As shown in the figure, the imputation performance is not very sensitive to the number of heads in the multi-head attention mechanism, except for categorical variable imputation in the Churn and Online datasets; overall, the DSAN shows the best performance when the *number of heads* is 4 for categorical variable imputation. Figure 9 shows the changes of imputation performance from varying the *denoising ratio*. This figure also shows that the imputation performance is not sensitive to the *denoising ratio* except for numeric variable imputation; for numerical variable imputation, a slight performance degradation is observed as the *denoising ratio* increases.

**Table 5.** The average categorical variable imputation performances.

| Dataset | Methods | Average Error Rate | Improvement |
|---|---|---|---|
| Adult | Mode | 0.4125 | +86.66% |
| | MissForest | 0.2288 | +3.53% |
| | MIDASpy | 0.5225 | +136.42% |
| | DSAN | 0.2210 | |
| Bank | Mode | 0.4080 | +55.68% |
| | MissForest | 0.2606 | −0.57% |
| | MIDASpy | 0.4402 | +68.02% |
| | DSAN | 0.2620 | |
| Online | Mode | 0.4180 | +20.31% |
| | MissForest | 0.3501 | +0.76% |
| | MIDASpy | 0.6596 | +89.81% |
| | DSAN | 0.3475 | |
| Churn | Mode | 0.4043 | +13.67% |
| | MissForest | 0.3648 | +2.57% |
| | MIDASpy | 0.5831 | +63.98% |
| | DSAN | 0.3556 | |

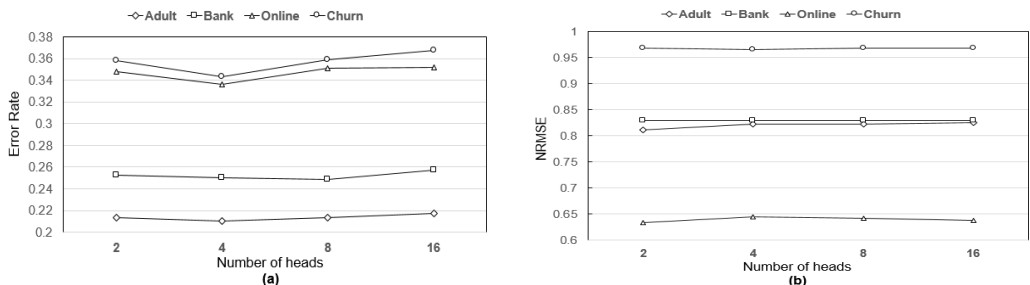

**Figure 8.** (**a**) Changes of *error rate* for categorical data imputation from varying the *number of heads* in the multi-head attention. (**b**) Changes of *NRMSE* for numeric data imputation from varying the *number of heads* in the multi-head attention.

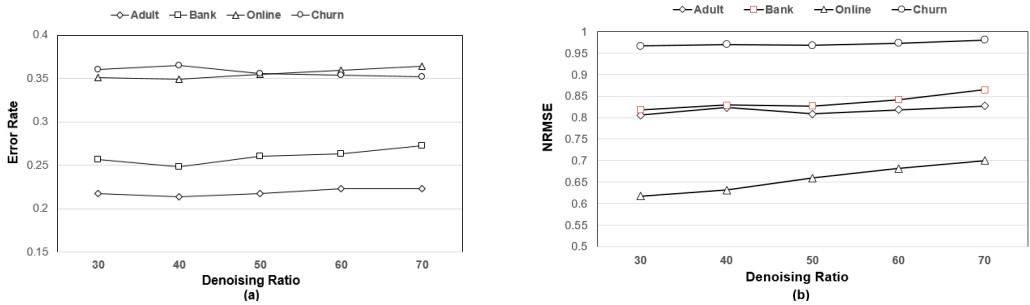

**Figure 9.** (**a**) Changes of *error rate* for categorical data imputation from varying the *denoising ratio*. (**b**) Changes of *NRMSE* for numeric data imputation from varying the *denoising ratio*.

*5.4. Downstream Task Performance*

To evaluate downstream task performance, we trained a binary classifier (using logistic regression) with an imputed dataset and evaluated classifier performance using 20% of the test dataset. All experimental datasets featured binary-labeled classes: 'Income' in Adult, 'Deposit' in Bank, 'Revenue' in Online, and 'Exited' in Churn.

Table 6 shows the downstream task performances using the AUC-ROC score as the evaluation metric; this reveals how well a classifier separates two classes in a dataset. A score closer to 1 indicates a better classifier. The original performance is that after training with the complete dataset but without generating missing values; thus, this is the upper bound of classifier performance. If the performance is similar to that of a classifier trained with the original dataset, the imputed and original datasets are of similar quality.

**Table 6.** Results of downstream task performance (AUC-ROC figures).

| Dataset | Methods | 5% Missing | 10% Missing | 15% Missing | 20% Missing |
|---|---|---|---|---|---|
| Adult | Original | | 0.9052 | | |
| | Mean/Mode | 0.9034 | 0.9015 | 0.8978 | 0.8962 |
| | MissForest | 0.9049 | 0.9049 | 0.9041 | 0.9036 |
| | DSAN | 0.9050 | 0.9050 | 0.9047 | 0.9045 |
| Bank | Original | | 0.9030 | | |
| | Mean/Mode | 0.9020 | 0.9002 | 0.8989 | 0.8962 |
| | MissForest | 0.9030 | 0.9028 | 0.9028 | 0.9019 |
| | DSAN | 0.9028 | 0.9029 | 0.9029 | 0.9020 |
| Online | Original | | 0.8945 | | |
| | Mean/Mode | 0.8889 | 0.8870 | 0.8796 | 0.8765 |
| | MissForest | 0.8952 | 0.8983 | 0.8980 | 0.8993 |
| | DSAN | 0.8951 | 0.8970 | 0.8964 | 0.8984 |
| Churn | Original | | 0.8325 | | |
| | Mean/Mode | 0.8324 | 0.8327 | 0.8308 | 0.8317 |
| | MissForest | 0.8322 | 0.8313 | 0.8298 | 0.8295 |
| | DSAN | 0.8323 | 0.8320 | 0.8316 | 0.8307 |

Table 7 shows the average downstream task performances. Compared to a classifier trained with the original dataset, the relative performance deteriorations ranged up to 0.1%. This shows that the DSAN-imputed and the original datasets were of similar quality. For the Online dataset, the performance improvement was 0.2% because some noise in the original data was resolved during imputation. In terms of mean/mode imputation, the relative performance improvements were 0.56% for the Adult, 0.37% for the Bank, and 1.55% for the Online datasets. For the MissForest dataset, the performance did not improve.

**Table 7.** Average downstream task performances.

| Dataset | Methods | Average | Improvement |
|---|---|---|---|
| Adult | Original | 0.9052 | −0.00% |
| | Mean/Mode | 0.8997 | +0.56% |
| | MissForest | 0.9044 | +0.05% |
| | DSAN | 0.9048 | |
| Bank | Original | 0.9030 | −0.00% |
| | Mean/Mode | 0.8993 | +0.37% |
| | MissForest | 0.9026 | +0.00% |
| | DSAN | 0.9026 | |
| Online | Original | 0.8945 | +0.20% |
| | Mean/Mode | 0.8830 | +1.55% |
| | MissForest | 0.8977 | −0.11% |
| | DSAN | 0.8967 | |
| Churn | Original | 0.8325 | −0.10% |
| | Mean/Mode | 0.8319 | −0.03% |
| | MissForest | 0.8307 | +0.11% |
| | DSAN | 0.8316 | |

*5.5. Interpreting Attention*

Unlike MLP-based models, which are difficult to interpret, some results afforded by *attention*-based models can be interpreted. The DSAN learns the interactions among each variable using a self-attention layer. It is possible to detect the features employed to make decisions. We employed the Adult dataset to explore this topic. This dataset contains individual annual incomes, and various relevant factors. The Adult data include 'relationship' and 'sex' variables. The 'relationship' variable represents a family relationship and can assume values such as 'Husband', 'Wife', and 'Unmarried'. If the 'relationship' variable has a 'Husband' value, the 'sex' variable must be 'Male'. This can serve as the basis for imputation of a value for 'sex' using the 'relationship' variable.

To confirm the learning ability of DSAN, we intend to take the *attention* weights from the self-attention layer in the feature representation module shown in Figure 3 (A) and to identify what features DSAN focuses on by visualizing the *attention* weights in the form of a heatmap. Figure 10 illustrates a visual representation of the attentional weights that reveal the DSAN decision-making process when imputing missing values. The relatively high weighted features can be considered to have contributed more to predicting missing values; that is, analyzing the *attention* weights of the missing value embedding vector can allow one to detect the highly contributing features for the MVI task.

Specifically, this figure shows the self-attention weights of 20 records when the 'relationship' variable had the 'Husband' value and the 'sex' variable predicted the substituted value 'Male' for the missing value. Each row is the *attention* weight of the 'sex:NULL' feature of the self-attention layer. It is apparent that the 'sex:NULL' features focus on the 'relationship:Husband' feature, indicating that the DSAN can capture the interactions among other variables when predicting substituted values.

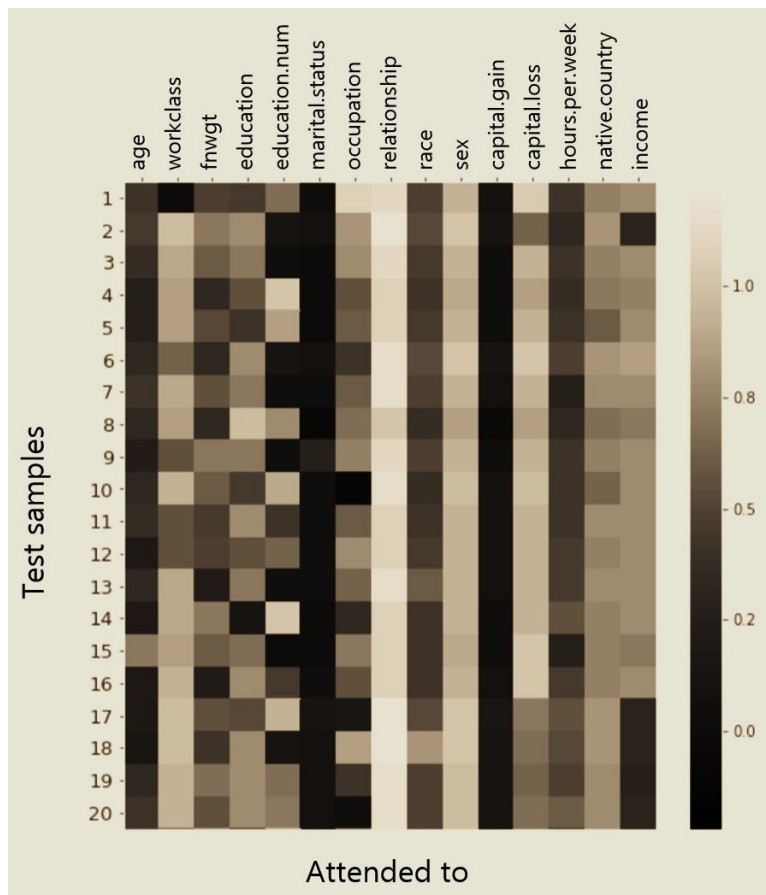

**Figure 10.** Self-attention weights of 20 records of the 'sex:NULL' feature when the 'relationship' variable had the 'Husband' value.

## 6. Conclusions

We proposed a novel, end-to-end imputational neural network termed the Denoising Self-Attention Network to solve the problem of missing values within mixed-attribute tabular data; our method includes a self-attention neural network architecture for effective representation learning for noisy data. To verify the model, we downloaded several real-world tabular datasets and evaluated two types of performance, of which one was imputational performance; that is, the *NRMSE* for numerical variables and the *error rate* for categorical variables were measured. The DSAN outperformed the other methods in terms of imputation of both numerical and categorical variables. We also explored downstream task performance. We trained a binary classifier using original and imputed datasets, and evaluated performance. A classifier trained with DSAN-imputed data performed similarly to a classifier trained with original data. Thus, the imputed dataset was of similar quality to the original dataset because the DSAN constructed a high-quality training dataset. As the number of records in tabular data to be imputed is greater, the imputation performance of DSAN increases. In addition, since the DSAN creates imputation models for all columns simultaneously via multi-task learning, its imputation process is much faster than that of the existing methods of generating an individual imputation model for each column.

However, the DSAN did not show sufficient performance improvement when imputing missing values for numerical variables. Moreover, the DSAN shows relatively low imputation performance for datasets with a small number of records. Thus, by refining the DSAN's architecture to specifically consider numerical variables and preventing degradation of some tasks in the multi-task learning, we will try to overcome these limitations. Basically, our proposed method relies on the MCAR type of missing data assumption even though MAR (Missing At Random) and MNAR (Missing Not At Random) types are more

common than the MCAR type. As another future study, we need to investigate whether the DSAN can be extended so as to be applied to the MAR and MNAR types.

**Author Contributions:** All authors made contributions to this work. H.-j.K. contributed to the organization of the research as well as the final manuscript preparation. D.-H.L. proposed the original idea, conducted data processing, and wrote the original draft of this work. Conceptualization, H.-j.K.; methodology, H.-j.K.; software, D.-H.L.; validation, H.-j.K.; formal analysis, D.-H.L. and H.-j.K.; investigation, H.-j.K.; resources, H.-j.K.; data curation, D.-H.L.; writing—original draft preparation, H.-j.K. and D.-H.L.; writing—review and editing, H.-j.K.; visualization, D.-H.L.; supervision, H.-j.K.; project administration, H.-j.K.; funding acquisition, H.-j.K. All authors have read and agreed to the published version of the manuscript.

**Funding:** This research was supported by Basic Science Research Program through the National Research Foundation of Korea (NRF) funded by the Ministry of Education (NRF-2022R1A2C1011937), and was also supported by the MSIT (Ministry of Science and ICT), Korea, under the ITRC (Information Technology Research Center) support program (IITP-2023-2018-08-01417) supervised by the IITP (Institute for Information & Communications Technology Planning & Evaluation).

**Institutional Review Board Statement:** Not applicable.

**Informed Consent Statement:** Not applicable.

**Data Availability Statement:** The data presented in this study are openly available at https://github.com/uos-dmlab/Structued-Data-Quality-Analysis.

**Conflicts of Interest:** The authors declare no conflict of interest. The funders had no role in the design of the study; in the collection, analyses, or interpretation of data; in the writing of the manuscript; or in the decision to publish the results.

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
