# Peer review of "A Self-Attention-Based Imputation Technique for Enhancing Tabular Data Quality"

_data, 2023_

Round 1

Reviewer 1 Report

In this paper, the authors propose a novel imputation neural network named DSAN, which can impute both categorical and numerical values.  The proposed methodology was tested on 4 datasets and compared with other techniques to evaluate its imputation capability. In addition, the authors also report a graphical representation to show the decision-making process of the algorithm. 

Overall, the paper is clearly written and contains sufficient details for good comprehension.  I think only a few minor changes are needed:

I suggest enriching the related work section. Some additional works should be added including not only neural network-based methods but also other approaches.  In particular, I recommend mentioning at least these works: 10.1016/j.knosys.2021.107114, 10.1109/TKDE.2018.2883103, 10.5441/002/edbt.2022.05, 10.48550/arXiv.1702.00820.

I also suggest the authors to improve the quality of the images, specifically Figure 3 and Figure 5 seem a bit too grainy. Furthermore, I suggest changing the colors of Figure 6 to improve the visualization. 

Finally, I suggest including future developments in the Conclusion section. 

Reviewer 2 Report

The author work on decision making using iputational neural network and their application

The author improve the paper by incorporating the following suggestion

1) Abstract should be improved by adding the novelty and motivation

2) The author should give detail description why they choose this method and what is the motivation 

3) The author should improve the English language and provide technaical word 

4) Reference list must be updated

Reviewer 3 Report

In this paper, a data imputation method for table data is proposed based on self-attention framework. The paper described the algorithm and conducted a serious test to prove its efficiency. However, several issues should be addressed or improved before the paper can be considered for publication.

1 The algorithm implementation and training computation complexity is not described in the paper which is necessary for readers.

2 The paper only compared the proposed method with MissForest a method proposed ten years ago. It is necessary to compare with some new method especially the deep learning based method to show the performance of the proposed algorithm.

3 According to the experiment results, there is not very much difference between the proposed method with MissForest. Just some minor improvement is achieved. What is the meaning of the study and how it can help with the data imputation? Is it necessary to use such a complex algorithm to just get such a small improvement?

4 the authors should supply an ablation study to show how the training data selection and hyperparamters  may affect the results. 

5 It is better to give some real examples to prove the proposed method is useful in some cases. Otherwise it is difficult for the readers to get when and why the proposed method is necessary.

Round 2

Reviewer 3 Report

The authors have revised the paper according to the comments from reviewers. It is suggested to accept the paper for publication.